# How Melatonin Affects Plant Growth and the Associated Microbiota

**DOI:** 10.3390/biology14040371

**Published:** 2025-04-03

**Authors:** Elisa Gamalero, Bernard R. Glick

**Affiliations:** 1Dipartimento di Scienze e Innovazione Tecnologica, Università del Piemonte Orientale, Viale T. Michel 11, 15121 Alessandria, Italy; 2Department of Biology, University of Waterloo, Waterloo, ON N2V, Canada

**Keywords:** plant growth promotion, plant stress relief, melatonin–plant interaction, melatonin–microbiome interaction

## Abstract

Melatonin is a hormone produced by Prokaryotes and Eukaryotes. In plants its role is mainly related to stress relief and plant development. Some soil bacteria synthesize melatonin, influencing melatonin levels in plants and affecting plant growth, stress resistance, and overall health. In turn, plant-derived melatonin affects the microbial community in the rhizosphere, creating a feedback loop between plants and soil microorganisms. While the role of melatonin in plant stress tolerance is well documented, only a small number of papers deal with the interaction of plants with soil bacteria able to produce this hormone or with its effects on soil microbiota. This paper aims to review the most recent literature on these topics in order to explore the interplay between melatonin, plant-beneficial bacteria, and plants. By shedding light on these interactions, we can develop innovative strategies to improve plant health and stress adaptation, leading to more sustainable agricultural systems.

## 1. Introduction to Melatonin

Melatonin (Figure 1) is a naturally occurring indolamine that is produced by both prokaryotes and eukaryotes. It was first found in 1958 as a secretion of bovine pineal glands [1]. Melatonin is best known as a brain hormone that regulates the sleep cycle in humans. In this regard, it is sold (over the counter) as an inexpensive sleep aid and as a means for international travelers to overcome jet lag and adjust to a new time zone following a long airplane flight. In addition, melatonin acts to combat oxidative stress by neutralizing free radicals and by promoting the gene expression of several antioxidant enzymes such as superoxide dismutase, catalase, glutathione peroxidase, and glutathione reductase [2,3]. Melatonin is also thought to regulate weight gain in humans, an activity that is based on the inhibitory effect of melatonin on the expression of the hormone leptin, which helps to maintain body weight [4].

Melatonin was first found in plants in 1995 [5], and since that time it has been shown to be present in many different plant species [6]. The chemical structure of melatonin is similar to the chemical structure of the phytohormone indole-3-acetic acid (Figure 1), with the biosynthetic pathways of these two compounds in plants both starting with the amino acid tryptophan [7]. It is believed that in all organisms the synthesis of melatonin involves four main steps that are catalyzed by six different enzymes [8]. However, melatonin biosynthesis diverged during evolution, leading to different pathways occurring in vertebrates, plants, and microorganisms. In plants, tryptophan is converted into tryptamine by a tryptophan decarboxylase. Then, serotonin is synthesized from tryptamine and catalyzed by tryptamine 5-hydroxylase. Starting from serotonin, at least three different biosynthetic pathways have been observed. The first one is based on the acetylation of serotonin to form *N*-acetylserotonin, which occurs in the chloroplast and is catalyzed by serotonin *N*-acetyltransferase. Then, the enzyme *N*-acetylserotonin methyltransferase converts *N*-acetyl-serotonin into melatonin via a methylation that occurs in the cytoplasm [9,10,11] (Figure 2).

A second pathway can drive melatonin synthesis when the plants are exposed to stress, inducing the accumulation of a high level of melatonin. In this case, *N*-acetylserotonin methyltransferase catalyzes the transformation of serotonin into 5-methoxytryptamine, and then produces melatonin by the action of serotonin *N*-acetyltransferase [12,13]. In *Arabidopsis thaliana* [14] and rice [15] the multifunctional enzyme caffeic acid O-methyltransferase, which is also responsible for lignin and flavonoid synthesis, behaves as the key component of a third pathway catalyzing the methylation of *N*-acetylserotonin in melatonin (Figure 2).

While it is well known that bacteria can produce melatonin in different amounts according to the considered strain, very little information on the melatonin biosynthesis pathways of these strains is available. In the two endophytes *Bacillus amyloliquefaciens* SB9 and *Pseudomonas fluorescens* RG11 the synthesis of melatonin occurs through 5-hydroxytryptophan, serotonin, and *N*-acetylserotonin intermediates. In these bacterial strains, tryptamine was not produced as an intermediate [16,17]. However, other studies revealed the production of tryptamine by diverse bacterial species such as *Lactococcus lactis*, *Leuconostoc* spp. [18], *Bacillus cereus* [19], *Hafnia alvei*, *Morganella morganii*, and *Klebsiella pneumoniae* [20]. Similarly, more recently the melatonin biosynthetic pathway tryptophan→tryptamine→serotonin→melatonin was found in *Bacillus safensis* (EH143) [21]. Moreover, two tryptophan decarboxylases were found in *Clostridium sporogenes* and *Ruminococcus gnavus* cultures [22].

Information on bacterial tryptophan-specific decarboxylase genes is rather limited [22] and no bacterial tryptophan hydroxylase gene has as yet been described. The lack of specificity of this enzyme, able to use other molecules besides tryptophan as a substrate, suggests a putative substrate promiscuity, which partly justifies the scarcity of papers focused on the functioning and role of the two enzymes [23].

It has been hypothesized that the initial role of melatonin was to behave as a free radical scavenger. Since melatonin is widely distributed among cyanobacteria and alpha Proteobacteria [24], it is reasonable to think that this compound has existed for a long time in biological history given that the ability to produce melatonin has been retained throughout the evolutionary process of all organisms [11]. Moreover, it has been speculated that melatonin biosynthesis appeared first in bacteria and, following the endosymbiotic theory formulated by Lynn Margulis [25], then spread to eukaryotes. Based on this theory, after ingestion alpha Proteobacteria become mitochondria, the phagocytosis of cyanobacteria evolves into chloroplasts, and both cellular organelles maintain the ability to synthesize melatonin. Consequently, all living organisms can produce melatonin, which plays an antioxidant role against free radicals from cellular respiration and photosynthesis in mitochondria (animals and plants) and chloroplasts (plants) [11]. Due to its involvement in the detoxification of free radicals, modulation of biological rhythms, and suppression of inflammation melatonin is considered a pleiotropic compound and the capability to synthesize it was transmitted to all forms of life because of its usefulness.

In plants, melatonin acts as an antioxidant that helps plants deal with both biotic and abiotic stress. Plant melatonin has also been reported to be involved in improving seed germination, fruit ripening, photosynthesis, biomass production, circadian rhythm, membrane integrity, root development, leaf senescence, osmoregulation, and stress modulation. As an example, Xiao et al. [26] demonstrated that at low concentrations (10–20 μM) melatonin promoted cotton seed germination mainly by increasing the antioxidant enzymes, while high concentrations (100–200 μM) inhibited germination. Melatonin concentrations can vary enormously in different plants, different stages of plant growth, different plant organs, and both the season and time of day that a plant is harvested. In fact, the content of melatonin in *Paeonia lactiflora* changes according to flower developmental stages [27]. The concentration of melatonin increased in the flower’s bud stage, decreased at the beginning of the bloom stage, peaked during the bloom stage, and again slowed down at the wither stage. The amount of melatonin in plant samples may be measured by radioimmunoassay, enzyme-linked immunosorbent assay, gas chromatography–mass spectrometry, and high-performance liquid chromatography with electrochemical detection, fluorescence detection, or mass spectroscopy. Various studies have reported a low of 0.02 pg/g fresh weight of kiwi fruit [28] and a high of 142,000 pg/g fresh weight of tomato leaves [29]. Melatonin is widely recognized as a plant growth regulator [7,30,31,32,33,34]. In this regard, melatonin has been shown to reduce oxidative stress, promote root growth and development, prevent leaf senescence, promote lateral root formation, stimulate gene expression of enzymes involved in photosynthesis, increase plant yield, and protect plants from phytopathogen attack. In addition to promoting plant growth, melatonin also influences the development of flowers and fruits. The work published by Liu et al. [35] demonstrated that melatonin can affect both the yield and quality of tomato. Tomato seeds soaked in a 0.1 mM melatonin solution generated plants that gave higher yields, measured as single-fruit weights; at the same time, tomato fruits contained more ascorbic acid, lycopene, and calcium. Similarly, plant irrigation with a nutrient solution containing melatonin induced significant enhancement of the contents of soluble solids, ascorbic acid, lycopene, citric acid, and P compared to control plants. The increase in the fruit size after melatonin treatment seems to be related to the modulation of the expression of genes involved in carbohydrate metabolism [36]. Moreover, melatonin has been implicated in the regulation of the circadian rhythm in plants.

In one recent study [37] the interactive effects of melatonin and salicylic acid on canola (*Brassica napus*) under drought stress conditions were elaborated. As expected, drought stress caused a reduction in canola shoot length, canola fresh and dry biomass, overall canola yield, photosynthetic rate, water potential, and osmotic potential. The plant endeavored to protect itself from the drought stress by increasing its level of free amino acids, soluble sugars, and various antioxidant enzymes such as catalase, peroxidase, superoxide dismutase, and ascorbate peroxidase. However, both seed priming (with 0.1 μM metformin and 0.5 mM salicylate) and foliar application (with the same levels of metformin and salicylate) with treatments of metformin and salicylate (applied both separately and together) decreased the negative effects of drought. This study showed that the synergistic application of metformin and salicylate significantly reduced the stress of canola plants from moderate drought conditions. When these compounds were tested separately, melatonin was slightly better than salicylate at reducing the effects of drought stress.

The effect of exogenous melatonin on plant growth in the presence of heavy metal contamination was investigated by Lv et al. [38]. Tobacco seedlings were sprayed with exogenous melatonin at different concentrations (0, 50, 100, 150, and 200 µmol/L) and cultivated in the presence of cadmium (100 µmol/L). The treatment with melatonin alleviated the heavy metal toxicity, with the best effect recorded with melatonin 200 µmol/L. Root length, plant biomass, chlorophyll content, plant height, and leaf length with this concentration of melatonin increased by 147.92%, 245.39%, 35.56%, 71.91%, 37.33%, and 307.41%, respectively, compared to seedlings exposed to Cd. In seedlings treated with melatonin at the highest concentration and subjected to abiotic stress malondialdehyde and Cd amounts were reduced by 34.30 and 64.63%, respectively, compared to melatonin untreated seedlings. The main mechanisms related to plant tolerance increase against abiotic stress was the mitigation of Cd-induced oxidative damage through the improvement of water uptake ability and the boost of antioxidant enzymes in cells. In fact, increases in peroxidase, superoxide dismutase, and glutathione by 334.01%, 140.39%, and 414.11%, respectively, were observed in seedings treated with 200 µmol/L of melatonin. Genes related to the production of melatonin and antioxidant enzyme, and to Cd transport, were modulated by exogenous melatonin and were involved in cadmium tolerance. This kind of study can provide valuable insights for developing effective environmental strategies to protect plants exposed to environmental stresses, while also establishing the theoretical basis for further research on the role of melatonin as a plant growth regulator involved in alleviating plant stress.

In this first section of this review, we described the discovery of melatonin, how it is synthesized in plants and bacteria, and its function in plants. The information provided is supported by extensive literature. Indeed, the growing interest in plant endogenous melatonin and the effects of exogenous melatonin on plant growth and health is evidenced by the publication of over 100 papers on these topics in the first three months of 2025 alone. However, the primary aim of this review is to expand knowledge on the latest findings from the scientific community regarding the interplay between melatonin, plant-beneficial bacteria, and plants, an area that, despite its significance, remains less explored in the literature compared to the broader research on melatonin’s role in plant physiology. In the following two sections, we will discuss (i) how melatonin produced by soil bacteria or applied exogenously, in combination with Plant Growth-Promoting Bacteria (PGPB), can enhance plant tolerance to environmental stress and (ii) how plant-derived melatonin can induce shifts in soil microbiota composition.

## 2. Melatonin, Bacteria, and Plant Stress Relief

Melatonin is mainly recognized for its role as a regulatory hormone in animals, but is increasingly being identified as a key player in plant physiology. Certain plant-associated bacteria can synthesize melatonin, suggesting a novel inter-kingdom mechanism by which these microbes could help plants to mitigate various types of stress. Moreover, several papers analyzed the effect of the combined treatment with exogenous melatonin and PGPB on the development of plants growing in stressful conditions. Bacterially produced or exogenously applied melatonin appears to modulate plant responses to a range of abiotic stressors—such as drought, salinity, UV radiation, and extreme temperatures (Figure 3)—by enhancing antioxidant defenses, influencing hormone signaling, and altering plant gene expression [16,17,21,39,40,41,42,43]. This section aims to review the recent literature about stress relief in plants by melatonin synthesized by rhizosphere bacteria or exogenous melatonin applied together with plant-beneficial bacteria.

Among root-associated bacteria, endophytes able to synthesize melatonin have received wide attention. To the best of our knowledge, the first manuscript reporting the synthesis of melatonin by an endophytic bacterium was in 2016 and focused on the strain *Bacillus amyloliquefaciens* SB9, isolated from the root tissue of the grape *Vitis labruscana* ‘Summer Black’ [16]. Once inoculated into *V. lambruscana* plantlets, this bacterial strain significantly increased several plant parameters including root length, plant height, biomass, and leaf chlorophyll content. To assess the effect of strain SB9 on plant growth under stressful conditions, salinity or drought stress was imposed on inoculated or uninoculated plantlets by adding NaCl solutions (60 or 120 mM) or PEG-6000 (10%). The data obtained indicated that strain SB9-inoculated plants exposed to stress showed an upregulation of the synthesis of melatonin and tryptamine, 5-hydroxytryptophan, serotonin, and *N*-acetylserotonin, melatonin intermediates. On the contrary, transcription of the tryptophan decarboxylase and serotonin *N*-acetyltransferase genes was reduced by plant inoculation with the bacterial strain. Finally, plant treatment with strain SB9 induced relief from the imposed stress by lowering the level of malondialdehyde and reactive oxygen species (H_2_O_2_ and O_2_^−^) in roots.

Similarly, *Pseudomonas fluorescens* RG11, isolated from surface-sterilized roots of grapevine cv. Red Globe, was selected for its ability to produce melatonin from tryptophan, especially during the exponential phase of growth. Plantlets of *Vitis vinifera* were inoculated with strain RG11 or subjected to salinity stress (80 mM NaCl). The presence of the bacterial strains in plantlets exposed to salt showed an increase in the endogenous amounts of 5-hydroxytryptophan, *N*-acetylserotonin, and melatonin, and reduced concentrations of tryptamine and serotonin. Moreover, inoculation with *P. fluorescens* RG11 decreased transcription of the tryptophan decarboxylase and serotonin *N*-acetyltransferase genes in the plant tissue compared to control plants, leading to a reduction in ROS levels in plants exposed to salt stress. Finally, to assess the possible influence of the grapevine cultivar on melatonin synthesis, the bacterial strain was inoculated on the root systems of three other cultivars (Riesling, Chardonnay, and Cabernet Sauvignon). The results obtained highlighted the fact that *P. fluorescens* RG11 can promote plant growth of grapevines belonging to the four cultivars considered and upregulates the melatonin level in plants cultivated in the presence of excess of salt [17].

Recently, Jofre et al. [39] explored the role and the effect of two PGPB strains, *Enterobacter* 64S1 and *Pseudomonas* 42P4, able to synthesize melatonin and promote the endogenous melatonin produced in *Arabidopsis thaliana* plants exposed to drought stress. The data obtained in this work demonstrated that inoculation with these bacterial strains enhanced the level of endogenous melatonin and plant growth by improving plant tolerance to the water deficiency and preventing oxidative damage. One year later, the same group demonstrated that inoculation of the strains S4S1 and 42P4 on tomato grown in drought stress promoted plant growth and increased the chlorophyll concentration and chlorophyll fluorescence parameters [35]. As already recorded in *A. thaliana*, the level of melatonin inside plant tissues increased after bacterial colonization. While the level of proline in tomato plants was enhanced in the presence of *Enterobacter* 64S1 and *Pseudomonas* 42P4, lipid peroxidation decreased, thus leading to improved tolerance of water deficit.

Recently, Kwon et al. [21] selected a bacterial strain (*Bacillus safensis* EH143) able to produce melatonin when incubated with a tomato plant. The bacterial strain synthesized melatonin following the pathway tryptophan→tryptamine→serotonin→melatonin and tolerated salinity (NaCl > 800 mM) and cadmium (3 mM). Once inoculated on soybean plants subjected, or not, to heavy metal and salinity stress, the bacterial strain EH143 accumulated a high amount of cadmium inside its cells, improved nutrient uptake, and reduced the level of Cd inside shoot tissues. Moreover, plants inoculated with strain *B. safensis* EH143 showed a lower ABA concentration and a higher amount of salicylic acid in the shoots compared to control levels, uninoculated salt-exposed, and uninoculated Cd-treated plants. Although both abiotic stresses boosted the expression of the gene ASMT3 involved in melatonin synthesis in the plants, no significant variation in the melatonin amount was detected in inoculated or uninoculated plants. However, the endogenous melatonin concentration in the roots and shoots was reduced in plants colonized by the bacterial strain.

Based on the idea of combining environmentally friendly biostimulants, several researchers have focused on the effect of treating plants with both PGPB and exogenous melatonin. For example, the application of the nitrogen-fixing bacterium *Azotobacter chroococcum*, an organism able to release high amounts of exopolysaccharide (EPS), was combined with different amounts of exogenous melatonin (0, 25, 50, and 100 µM) and added to faba bean plants exposed to salt stress [41]. While salinity reduced faba bean growth, inoculation with the bacterial strain and/or the phytohormone promoted plant growth and increased plant yield by enhancing the concentration of macronutrients (N, P, K). Moreover, plants treated with *A. chroococcum* or melatonin mitigated the adverse effects of salinity stress by enhancing the level of photosynthetic pigments (chlorophyll a and b and carotenoid content) and proline content. Overall, the data presented in this paper demonstrated that both melatonin and *A. chroococcum* behave as biostimulants in faba beans grown under salt stress. Moreover, the combined application of the bacterial strain with melatonin led to an additive effect and induced the highest level of plant growth promotion.

In another study [42], a bacterial endophyte, *Bacillus safensis* BTL5 isolated from the perennial plant *Ocimum tenuiflorum*, a bacterium from a rice rhizosphere (*Brevibacterium frigoritolerans* W19), and melatonin were used to assess their effects on plant stress tolerance to salinity. Rice seeds were treated with each bacterial strain and/or melatonin (20 ppm) in the presence or absence of a high level of salt stress (200 mM NaCl). Inoculation of plants exposed to salinity with each of the bacterial strains increased the concentration of chlorophyll, proline, phenylalanine ammonia-lyase, catalase, superoxide dismutase, polyphenol oxidase, and plant growth parameters. When melatonin was added to bacterized plants synergistic effects on plants compared to all other treatments were observed, especially on root and shoot dry weight, shoot length, and chlorophyll content. Moreover, the data showed the highest levels of phenylalanine ammonia-lyase, the antioxidant enzyme catalase, and proline in plants inoculated with bacterial strains and then sprayed with melatonin.

The efficacy of PGPB together with exogenous melatonin on plant development was assessed under water deficit conditions by Imran et al. [43], who investigated the effects of the phytohormone application together with the bacterium *Lysinibacillus fusiformis* PLT16 on soybean plants. In this work special attention was given to hormonal, antioxidant, physiological, and molecular regulation in drought stress conditions. Strain PLT16 was characterized for its plant-beneficial traits and found to be able to produce exopolysaccharide, siderophore, and auxin and to tolerate water deficit. Strain *L. fusiformis* PLT16 was then used to inoculate soybean plants with or without the addition of melatonin and then subjected to drought stress. While the abiotic stress reduced photosynthetic efficiency, increased the ROS level, and decreased plant growth, the combination of the bacterial strain and melatonin favored plant development and increased the level of photosynthesis pigments (chlorophyll a and b and carotenoids). The positive effects of the combined treatment of bacteria and melatonin was apparently related to the reduction in the hydrogen peroxide, superoxide anion, and malondialdehyde levels and to the increase in antioxidant activities. This modulation of ROS levels led to a reduction in abscisic acid and an enhancement of jasmonic and salicylic acids, both involved in drought stress relief. In addition, an increase in endogenous melatonin content and improved nutrient uptake was observed in soybean plants treated with strain PLT16 and melatonin under both optimal and drought conditions.

Altogether, these results throw light on the interplay among melatonin, PGPB, and stress mitigation, showing the potential of both bacterial-synthesized and exogenous melatonin in enhancing plant resilience to various abiotic stresses such as drought, salinity, and heavy metal contamination. The often-observed synergistic effects of melatonin and PGPB not only boost plant growth but also modulate key physiological processes, including antioxidant defense systems, hormone regulation, and gene expression. Moreover, the ability of PGPB to synthesize melatonin demonstrates that these microbes can directly influence plant stress responses. Integrating melatonin and PGPB into agricultural practices could be a sustainable and effective strategy for improving crop resilience, optimizing yields, and reducing the introduction of chemical inputs in agroecosystems.

## 3. Melatonin and the Soil Microbiome

While its effects on human physiology are well documented, research on the effects of melatonin on bacteria is only just now developing significantly, expanding from its initial human clinical applications to a broader environmental context. Initially, studies of melatonin focused on its role in human health, particularly its antioxidant and regulatory effects. To the best of our knowledge, the first paper reporting an effect of melatonin on bacteria was published in 2000 [44]; this work showed that a relatively low concentration (1 mM) of melatonin increased the hydrophobicity of the cell surface of the human pathogen *Neisseria meningitidis* (the Gram-negative bacterium that causes meningitis), possibly decreasing its pathogenicity. Subsequently, Tekbas et al. [45] investigated the antibiotic effect of melatonin on both Gram-negative and -positive multidrug-resistant bacteria such as *Staphylococcus aureus*, carbapenem-resistant *Pseudomonas aeruginosa*, and *Acinetobacter baumannii*. After 24 h of incubation, the minimal inhibitory concentration (MIC) for melatonin was 250 µg/mL for Gram-positive bacteria, and 125 µg/mL for Gram-negative ones. Surprisingly, after 48 h of incubation MIC values decreased to 125 and 31.25 µg/mL for Gram-positive and Gram-negative bacterial strains, respectively, suggesting that melatonin modulates the replication of bacterial cells. Starting from these initial studies, many other papers reported the inhibitory effect of melatonin on human pathogens (for a review see [46]). While the initial focus of this research was an exploration of melatonin’s influence on pure bacterial strains, as a consequence of the development of new molecular tools the attention of the scientific world has turned towards more complex microbial communities.

Emerging research suggests that melatonin produced by bacteria can affect plant tolerance and health; in turn, melatonin synthesized by plants can influence the soil and rhizosphere microbiome. Altogether, melatonin can improve nutrient utilization and photosynthetic activity, increase plant tolerance to environmental stresses (both biotic and abiotic), and trigger the plant’s antioxidant capabilities. All these positive effects on plants lead to plant growth promotion, an increase in plant productivity, and an improvement of plant health. All these aspects are consistent with the directions and goals of sustainable agriculture and fit well with the ’One Health’ approach and the Goals of the 2030 Agenda, especially promoting sustainable agriculture (Zero hunger, Goal 2), combating climate change (Climate change, Goal 13), and preserving life on land (Life on land, Goal 15).

Even though there is only limited literature [47,48,49,50,51,52,53] dealing with the effects of melatonin on soil bacteria, reflecting the fact that this field is in its infancy, below we have discussed some of the key manuscripts that address this issue.

Madigan et al. [47] studied the effect of abiotic stress (cadmium or salt) on three agricultural soils subjected to different management regimens and cultivated crops (pasture, canola, and wheat) treated or not with melatonin. While melatonin did not affect the fungal alpha diversity, both an increase and a reduction in alpha diversity occurred in the bacterial community depending upon the considered soil. On the contrary, the bacterial and fungal microbiota changed following melatonin treatment or stress establishment: while a decrease in OTU (Operational Taxonomic Unit) abundance and evenness was observed in the bacterial community, an increase in the two indices was observed in the fungal community. High amounts of melatonin induced an increase or decrease in the alpha diversity of the bacterial community according to the considered soil, while the fungal community was modified by melatonin only in soil cultivated with wheat.

Additional research explored how melatonin influences plant performance and microbial community composition under specific stress conditions, such as drought. To explore the roles of melatonin on plant performance under drought stress and on rhizosphere bacterial and fungal communities, barley plants were subjected to drought and/or melatonin treatment [48]. Melatonin application yielded an improvement in the performance of plants grown under water deficit. This effect was related to a lower accumulation of H_2_O_2_ and an increased amount of glycine betaine in leaves. Moreover, the hormone treatment led to the increased expression of carbohydrate metabolism enzymes and the activation of antioxidant enzymes. Focusing on the microbiota, under drought stress rhizosphere treatment with melatonin reduced the bacterial biodiversity, while the fungal biodiversity did not change. However, both the hormone and the abiotic stress induced variations in the beta diversity of both the bacterial and fungal communities. In more detail, five phyla (*Actinobacteria*, *Proteobacteria*, *Acidobacteria*, *Chloroflexi*, and *Gemmatimonadetes*) were recognized as dominant in the samples. While water deficit increased the frequency of *Actinobacteria*, the amount of *Proteobacteria* was reduced. Plant treatment with melatonin induced the opposite effect, while no variation in the abundance of *Chloroflexi* was observed. Among fungi, *Ascomycota*, *Mortierellomycota*, *Basidiomycota*, and *Olpidiomycota* were dominant in the barley rhizosphere, with *Ascomycota* being the most frequent. *Ascomycota* responded to the abiotic stress by increasing their density, with melatonin further amplifying this effect. Drought significantly enriched the level of *Ascomycota* in the barley rhizosphere, and rhizospheric melatonin application further amplified this effect. On the other hand, the density of *Mortierellomycota* and *Basidiomycota* was reduced by both melatonin and drought compared to untreated controls.

Complementing these drought-related insights, another study examined the combined effects of melatonin and nutrient supplementation on plant growth and microbial diversity in open-field conditions. In this study, Xiao et al. [49] examined the effect of melatonin and urea application, alone or together, on soybean plant leaves. While foliar spray with urea (3 kg ha^−1^) improved plant growth, melatonin did not affect the plant productivity; the combination of the two induced a yield decrease compared to the treatment with just the urea nitrogen source. Moreover, the alpha diversity of the bacterial community remained unaffected following either of the urea or melatonin treatment. On the other hand, melatonin induced shifts in microbial community structure, whereas the combination melatonin plus urea did not. Both melatonin and urea modulated the abundance of specific taxa, with melatonin having a greater effect than urea.

In addition to its role in nutrient management, melatonin’s potential to enhance tolerance to environmental stresses, such as waterlogging, has also been investigated in controlled experiments [50]. Melatonin was added to waterlogged apple plants grown in pots. The main result was that melatonin increased tolerance to waterlogging by enhancing both the plant’s photosynthesis and antioxidant capability. Nutrient utilization (especially nitrogen) was improved thanks to the upregulation of the *nifD* and *nifK* genes. In addition, the apple plants’ soil microbiome was analyzed during the waterlogging treatment in the presence of melatonin to assess the effects of melatonin on the microbiome diversity during the stress relief. In fact, the bacterial diversity in melatonin-treated plants was lower than that observed in waterlogged plants at time 0 and after 14, 21, and 28 days. While the dominant phyla in all samples were *Proteobacteria*, *Actinobacteriota*, *Acidobacteria*, *Gemmatimonadota*, and *Chloroflexi*, the number of reads corresponding to *Proteobacteria* on day 7, 14, 21, and 28 was higher in the melatonin-treated waterlogged plants than in the waterlogged plants without melatonin. Similarly, at time 0 and after 14, 21, and 28 days *Actinobacteriota* numbers were higher in the melatonin-treated waterlogged plants than in the waterlogged plants without melatonin. Looking at the genera, bacteria belonging to *Azoarcus* and *Pseudomonas* were higher in melatonin-treated plants than in waterlogged plants without melatonin in all time periods considered. The frequency of the genera *Nocardioides*, *Anaeromyxobacter*, and *Citrifermentans* was higher in melatonin-treated plants than in waterlogged plants without melatonin in some of the sampling days. According to the results obtained, the authors concluded that melatonin positively regulates physiological functions in plants as well as the structure and function of the microbial community. Thus, melatonin promoted the recovery of apple plants following waterlogging stress.

A very recent study [51] investigated the effects of melatonin on the physiological and biochemical features of *Diospyros lotus* (date-plum) grown under water deficit conditions and during stress recovery. Through a comprehensive multi-omics approach, the manuscript provides an exhaustive description of the effects of melatonin on plant gene expression, key metabolic pathways, differentially synthesized molecules, and the composition of the associated microbiome. The experiment included control seedlings (CK, 70–80% soil water content), seedlings subject to water deficit (D, 40–50% water content) for one week, and seedlings desiccated for seven days and rewatered for one week with water (DR) or melatonin (MR). The metabolic pathways of sugars and flavonoids and the level of plant-synthesized ROS, especially catalase, were enhanced by melatonin, leading to a more efficient recovery from the abiotic stress. Looking at the microbiota, 10,269 OTUs were detected and the bacterial abundance in soil subjected to water deficit was higher than in the DR and MR treatments. Although *Proteobacteria* and *Acidobacteria* were the dominant phyla in all treatments, the genera in the DR and MR rhizospheres were significantly different from those observed in the CK and D treatments. The genera *Skermanella*, *Nocardioides*, *Ramlibacter Arthrobacter*, *Citrifermentans*, *Mesorhizobium*, *Aeromicrobium*, and *Bdellovibrio* were differentially enriched in the rhizosphere of DR- and MR-treated plants. Moreover, melatonin induced an increase in the members of the genera *Pseudomonas*, *Bacillus*, *Skermanella*, *Ralstonia*, *Rhodococcus*, *Nocardioides*, *Marmoricola*, and *Ramlibacter*, which are potentially beneficial microorganisms able to support the growth of the plant during the recovery period.

In addition to the epiphytic rhizosphere bacterial community, the endophytic bacterial community can also be affected by melatonin. Recently, Ma and colleagues [52] investigated the effect of melatonin on the phloridzin (a glucoside of phloretin: Figure 4) content on the endophytic microbiome in apple trees affected by apple replant disease. Apple replant disease is spread worldwide and leads to a significant growth inhibition, as well as a severe reduction in productivity and fruit quality following the replanting of apple trees on the same site [53,54]. Phloridzin is a phenolic acid involved in the regulation of apple growth and resistance to stress [55]. The high content of this molecule in soil can lead to increased risk of infection by phytopathogens. The *in vivo* experiments (pot and open-field conditions) were based on the hypothesis that melatonin can reduce apple replant disease (ARD) through the modulation of the phloridzin content at the root level and thereby induce shifts in the endophytic microbial community. Plant treatment with 200 μM melatonin increased the alpha diversity of the bacterial endophyte community while reducing the alpha diversity of the endophytic fungi. *Fusarium*,* Proteobacteria*,* Actinobacteria*, and *Chloroflexi* were the dominant endophytic bacteria while *Glomeromycota*, Ascomycota, and Basidiomycota were the dominant endophytic fungi. Among bacterial endophytes *Streptomyces*, *Steroidobacter*, *Rhodomicrobium*, *Verrucosispora*, and *Ramlibacter* were enriched by phloridzin, while *Lysobacter* and *Azovibrio* increased after phloridzin and melatonin treatment. *Podospora* and *Septoglomus* increased after the application of phloridzin and melatonin while *Talaromyces* was enriched by phloridzin. Moreover, melatonin reduced the inhibitory effect of ARD on apple plant growth and the amount of phloridzin in roots. An effect of melatonin was also observed on leaf antioxidant enzyme activities (superoxide dismutase, peroxidase, catalase, and ascorbate peroxidase) that were increased by the hormone. Overall, these results demonstrate that an improvement of the rhizosphere environment mediated by melatonin was associated with modification of the endophytic microbiota community, and a reduction in phloridzin levels in the rhizosphere soil and roots.

The research on melatonin’s effects on bacteria underscores its significance in environmental microbiology. These studies demonstrate that melatonin can influence microbial communities’ dynamics, showcasing its potential to enhance plant health and resilience against abiotic stresses. Boosting the diversity of beneficial bacterial genera both in the rhizosphere and in plant tissues, melatonin’s role as a plant growth regulator extends far beyond its traditional antioxidant properties. Notwithstanding the limited number of studies to date, the potential for growth in this research area is substantial. In fact, further exploration into how melatonin can be harnessed in agricultural practices to improve crop yield and sustainability, especially under stressed conditions, will provide a deeper understanding of the interactions between melatonin, plants, and microbial communities. This is especially relevant in the context of global challenges related to climate change, sustainable agriculture, and food security, where innovative strategies to improve plant stress tolerance and promote healthier soils are crucial.

## 4. Conclusions

With an ever-increasing world population, there is an urgent need to dramatically increase global agricultural productivity. With worldwide climate change as an unfortunate limitation on traditional agriculture in many environs, how we feed the world’s people requires thoughtful and innovative approaches. One such approach is the widespread employment of PGPB as a mainstay of agricultural practice instead of chemicals. However, before this can become a reality it is first necessary to understand how plants, soil, mycorrhizae, and PGPB interact with one another on a fundamental basis. This includes increasing our understanding of the key role that melatonin plays in promoting plant growth. In this regard, melatonin produced by bacteria and plants is not only a free radical scavenger, but it can also act to modify the soil and the rhizosphere microbiome, altering both the composition and biochemistry of these microbes, thereby protecting plants from stress in a multiplicity of ways.

## Figures and Tables

**Figure 1 biology-14-00371-f001:**
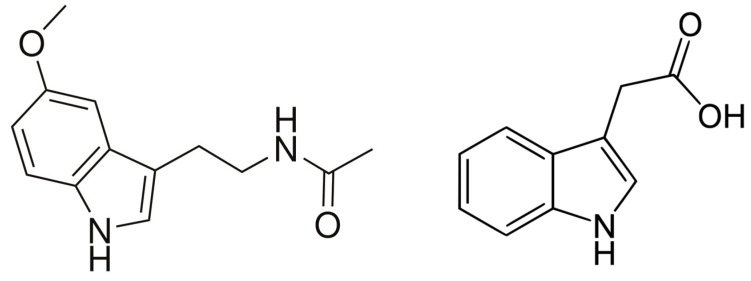
The chemical structure of melatonin (**left**) and indole-3-acetic acid (**right**).

**Figure 2 biology-14-00371-f002:**
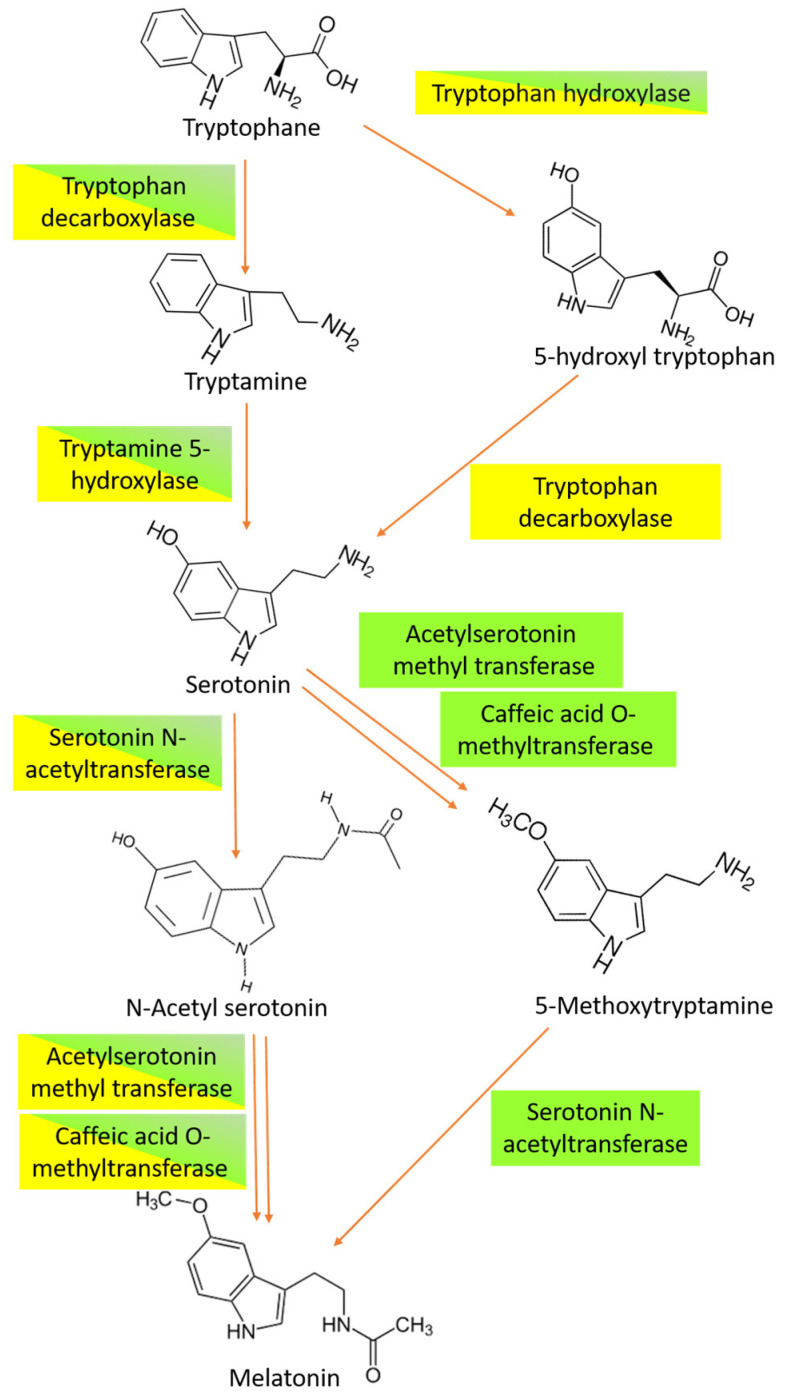
Melatonin metabolic pathways in plants and prokaryotes. The names of plant enzymes are given in a green box, while those of prokaryotic origin are in a yellow box. Enzymes common to plants and prokaryotes are shown in a yellow and green box.

**Figure 3 biology-14-00371-f003:**
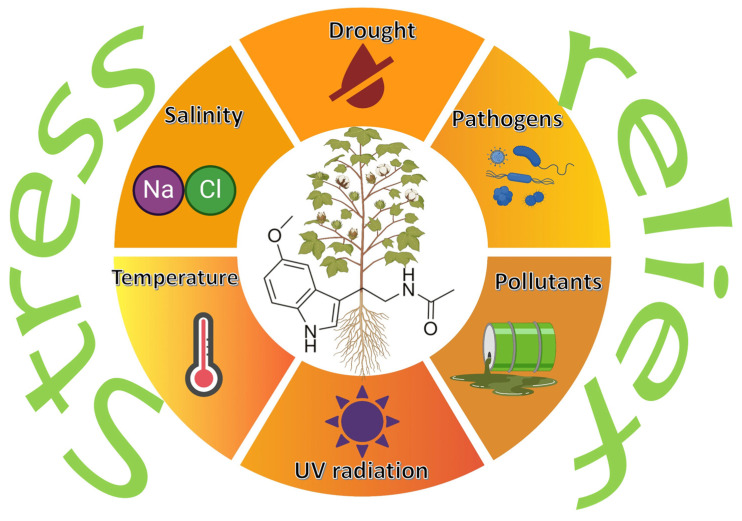
Role of melatonin in plant stress mitigation. This figure was created with BioRender.com Created in BioRender. Gamalero, E. (2025) https://BioRender.com/8wam8um (accessed on 31 March 2025).

**Figure 4 biology-14-00371-f004:**
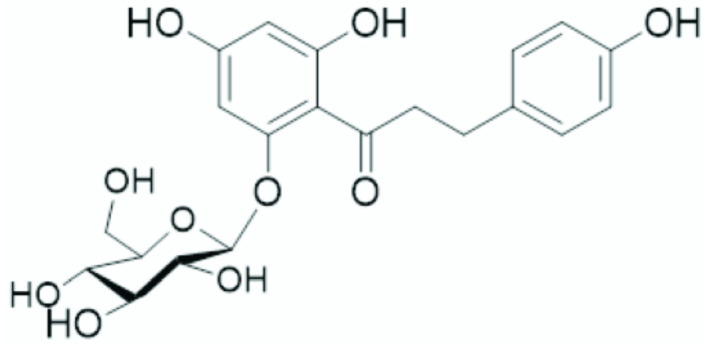
The chemical structure of phloridzin.

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
