# Peer review of "How Melatonin Affects Plant Growth and the Associated Microbiota"

_biology, 2025, doi:10.3390/biology14040371_

Round 1
Reviewer 1 Report
Comments and Suggestions for Authors
Dear authors;
Thank you for your fluent writing and your figures, but the emptiness of a few tables is clearly felt. Below I provide my commentaries.
- First, there are many articles that publish in 2025 and other years about melatonin effect on plant’s grows and …, pleas add them in your manuscript.
Introduction:
- In line 77 you said “there is some difference between Melatonin and phytohormone indole-3-acetic acid”. It is better to show both structures in figure 1 and compared them.
- It is valuable if you add a table about melatonin concentration in different plants for line 144-145.
- Melatonin, bacteria and plant stress relief
- There are no any references in paragraph 1, please add it. Also, it is not necessary to write “that will be listed here below” in line 171. We will read the text and find them.
- It is better to compared different result with each other or at least prepared a table to show various achievements and their results at a glance in all sections.
- Sometimes, it is better to say the author’s name instead of repeat “In another study”
- Melatonin and the soil microbiome
- Line 319-332, please add the reference.
Conclusions:
- Why you write a reference in conclusions? It is inappropriate self-citations, please remove it or add it in the main text.
Author Response
Thank you for your fluent writing and your figures, but the emptiness of a few tables is clearly felt. Below I provide my commentaries.
- First, there are many articles that publish in 2025 and other years about melatonin effect on plant’s grows and …, pleas add them in your manuscript.
A: As stated in the text, the interest on plant endogenous melatonin and on the effects induced by exogenous melatonin on plant growth and health is demonstrated by the fact that in the first three months of 2025 more than 100 additional manuscripts have been published on these topics. However, it is also true that the focus of this review is not the effect of melatonin on plant growth per se but it is on 1) the effects of plant melatonin on microbial communities and 2) the impact of bacterial melatonin on plant growth. Therefore, in our opinion, adding to the text the literature on the effect of melatonin on plant growth is somewhat outside of the scope of this review. Nevertheless, we have now explained in detail the effect of exogenous melatonin on tobacco seedling exposed to cadmium by using the paper by Lv et al., published in 2025.
Introduction:
- In line 77 you said “there is some difference between Melatonin and phytohormone indole-3-acetic acid”. It is better to show both structures in figure 1 and compared them.
A: We added to figure 1 the chemical structure of indole-3-acetic acid in order to allow the suggested comparison.
- It is valuable if you add a table about melatonin concentration in different plants for line 144-145.
- A: The content of melatonin in plants is highly variable according to plant species, stages of plant development, plant organs, and both the season and time of day that a plant is harvested. In addition, in flowers the amount of melatonin changes according to the light. While we cited the content of melatonin in the descripton of several papers, we don’t think a table would be more informative for the reader.
2.Melatonin, bacteria and plant stress relief
- There are no any references in paragraph 1, please add it. Also, it is not necessary to write “that will be listed here below” in line 171. We will read the text and find them.
A: The paragraph has been modied as suggested.
- It is better to compared different result with each other or at least prepared a table to show various achievements and their results at a glance in all sections.
A: We think this information is well covered in the text and adding a Table reporting the same informations given in the text would be redundant
- Sometimes, it is better to say the author’s name instead of repeat “In another study”
A: Done, as suggested.
3.Melatonin and the soil microbiome
- Line 319-332, please add the reference.
A: the paragraph has been written by us, so we don’t think it is necessary to add a reference.
Conclusions:
- Why you write a reference in conclusions? It is inappropriate self-citations, please remove it or add it in the main text
A: As suggested, the reference has been deleted from the conclusions section.
Reviewer 2 Report
Comments and Suggestions for Authors
General comments: This review comprehensively explores the dual role of melatonin in plant-microbe interactions, emphasizing its potential in sustainable agriculture. However, there are major problems in manuscript writing and content organization.
Specific comments:
1. Manuscript writing shares certain characteristics with prose. This is mainly because the author is uncertain about the perspective their manuscript will offer to the academic community and the knowledge gap it will fill. After carefully reading the article, it is evident that the author highlights the limited research on the impact of melatonin on soil bacteria.
2. In lines 25-26, "plant growth" should not be isolated.
3. The keyword is highly repetitive,and I recommended to modify.
4. In line 104-106, please reorganize the sentence.
5. Line 108-112, the meaning of the sentence is unclear.
6. Line 116-120, the introduction of microbial metabolic pathways is also somewhat confusing, and there is no center of expression.
7. Line 123-125, Please add references.
8. Line 146-165, The second paragraph is only the case of the first paragraph, why is it divided into two paragraphs?
9. Line 259,272,Repeat a sentence of In another study, Which can cause aesthetic fatigue. In addition, these paragraphs are merely a summary of the academic literature.
10. Line 316-318, the expression here reveals an unexpected discovery, but the paragraph is over.
11. Line 319-329, this section on health seems somewhat disconnected from the manuscript's main content and appears a bit of a stretch.
12. I think this manuscript needs to reconstruct the logic framework of the whole text. In fact, the contents of the latter two parts have already appeared in the first part, which leads to repeated cases, cross - citation, duplication and confusion in the literature. It is very important that the reference should be read, understood, and integrated by the authors to form a review after increasing their own cognition, rather than a large number of repeated specific work content of a study in the review.
Author Response
General comments: This review comprehensively explores the dual role of melatonin in plant-microbe interactions, emphasizing its potential in sustainable agriculture. However, there are major problems in manuscript writing and content organization.
Specific comments:
1. Manuscript writing shares certain characteristics with prose. This is mainly because the author is uncertain about the perspective their manuscript will offer to the academic community and the knowledge gap it will fill. After carefully reading the article, it is evident that the author highlights the limited research on the impact of melatonin on soil bacteria.
2. In lines 25-26, "plant growth" should not be isolated.
A: Done, as suggested.
3. The keyword is highly repetitive,and I recommended to modify.
A: Done, as suggested.
4. In line 104-106, please reorganize the sentence.
A: Done, as suggested.
- Line 108-112, the meaning of the sentence is unclear.
A: Done, as suggested.
- Line 116-120, the introduction of microbial metabolic pathways is also somewhat confusing, and there is no center of expression.
A: Done , as suggested.
7. Line 123-125, Please add references.
A: Done, as suggested.
8. Line 146-165, The second paragraph is only the case of the first paragraph, why is it divided into two paragraphs?
A: Done. As suggested.
9. Line 259,272,Repeat a sentence of In another study, Which can cause aesthetic fatigue. In addition, these paragraphs are merely a summary of the academic literature.
A: As suggested, we have modified the text accordingly.
10. Line 316-318, the expression here reveals an unexpected discovery, but the paragraph is over.
A:We agree with the reviewer and we added a sentence to explain the main results obtained in the paper.
11. Line 319-329, this section on health seems somewhat disconnected from the manuscript's main content and appears a bit of a stretch.
A: Although we understand the opinion of the reviewer, it should be taken into consideration that in this section we are briefly telling the history of the main discovery of melatonin. Thus, it is necessary to maintain this paragraph.
I think this manuscript needs to reconstruct the logic framework of the whole text. In fact, the contents of the latter two parts have already appeared in the first part, which leads to repeated cases, cross - citation, duplication and confusion in the literature. It is very important that the reference should be read, understood, and integrated by the authors to form a review after increasing their own cognition, rather than a large number of repeated specific work content of a study in the review.
A: As stated in the manuscript, while the literature regarding the role of melatonin in plants is quite abundant, the number of papers dealing with melatonin synthesized by soil bacteria or the effects of plant melatonin on the microbiota is very low. Thus, we were forced to use what literature was available to us and this led to the repetition of some references in the text.
Reviewer 3 Report
Comments and Suggestions for Authors
Melatonin is a naturally occurring indolamine that is produced by both prokaryotes and eukaryotes. It was first found in 1958 as a secretion of bovine pineal glands. In plants, melatonin acts as an antioxidant that helps plants deal with both biotic and abiotic stress. In the MS, the mechanism of melatonin promoting root growth and development, preventing leaf senescence, flowering and fruit ripening, lateral root formation, and stimulating gene expression of en-zymes involved in photosynthesis, and protecting plants from phytopathogen attack, were all discussed. It is useful for the application of melatonin in agriculture.
This paper reviewed the most recent literature on these topics in order to explore the interplay between melatonin, plant-beneficial bacteria, and soil microbiota. The role of melatonin in plant stress tolerance is well-documented, the interaction of plants with soil bacteria able to produce this hormone or with its effects on soil microbiota.
Additionally, up to now, the sale volume in agriculture and price, and the application prospect should be introduced.
Author Response
Melatonin is a naturally occurring indolamine that is produced by both prokaryotes and eukaryotes. It was first found in 1958 as a secretion of bovine pineal glands. In plants, melatonin acts as an antioxidant that helps plants deal with both biotic and abiotic stress. In the MS, the mechanism of melatonin promoting root growth and development, preventing leaf senescence, flowering and fruit ripening, lateral root formation, and stimulating gene expression of en-zymes involved in photosynthesis, and protecting plants from phytopathogen attack, were all discussed. It is useful for the application of melatonin in agriculture.
This paper reviewed the most recent literature on these topics in order to explore the interplay between melatonin, plant-beneficial bacteria, and soil microbiota. The role of melatonin in plant stress tolerance is well-documented, the interaction of plants with soil bacteria able to produce this hormone or with its effects on soil microbiota.
Additionally, up to now, the sale volume in agriculture and price, and the application prospect should be introduced.
A: We thank the reviewer for these positive comments .
Reviewer 4 Report
Comments and Suggestions for Authors
The manuscript provides a comprehensive review of melatonin's role in plant growth, stress tolerance, and its interactions with soil microbiota. The authors have effectively highlighted the potential of melatonin as a biostimulant and its implications for sustainable agriculture. However, there are areas where clarity, depth, and scientific rigor could be improved.
Line 10-21: Simple summary is almost similar to the abstract. Make it more concise.
Lines 62-63: Include a brief mention of the evolutionary significance of melatonin across different kingdoms.
Line 146-154: The authors mentioned that melatonin acts as a plant growth regulator, but it would be beneficial to provide more specific examples of how melatonin influences different growth stages (e.g., seed germination, flowering, fruit ripening) and the underlying molecular mechanisms involved.
Lines 381-385: The authors briefly mentioned melatonin's role in mitigating waterlogging stress, it does not adequately explore its effects on heavy metal stress.
Lines 361-367: The discussion on fungal communities focuses primarily on Ascomycota, neglecting other phyla like Basidiomycota and Mortierellomycota.
Author Response
Reviewer 4
The manuscript provides a comprehensive review of melatonin's role in plant growth, stress tolerance, and its interactions with soil microbiota. The authors have effectively highlighted the potential of melatonin as a biostimulant and its implications for sustainable agriculture. However, there are areas where clarity, depth, and scientific rigor could be improved.
Line 10-21: Simple summary is almost similar to the abstract. Make it more concise.
A: As suggested, the simple summary has been modified accordingly.
Lines 62-63: Include a brief mention of the evolutionary significance of melatonin across different kingdoms.
A: As suggested, a sentence has been added, i.e. lines 135-138.
Line 146-154: The authors mentioned that melatonin acts as a plant growth regulator, but it would be beneficial to provide more specific examples of how melatonin influences different growth stages (e.g., seed germination, flowering, fruit ripening) and the underlying molecular mechanisms involved.
A:As suggested, we have now discussed these topics by introducing the works published by Zhao D, Wang R, Liu D, Wu Y, Sun J, Tao J (2018) Melatonin and expression of tryptophan decarboxylase gene (TDC) in herbaceous peony (Paeonia lactiflora Pall.) flowers. Molecules 23:1164
Liu JL, Zhang RM, Sun YK, Liu ZY, Jin W, Sun Y (2016) The beneficial effects of exogenous melatonin on tomato fruit properties. Sci Hortic 207:14–20
Xiao S, Liu L, Wang H, Li D, Bai Z, Zhang Y, Sun H, Zhang K, Li C. Exogenous melatonin accelerates seed germination in cotton (Gossypium hirsutum L.). PLoS One. 2019 Jun 25;14(6):e0216575. doi: 10.1371/journal.pone.0216575. PMID: 31237880; PMCID: PMC6592504.
Arnao MB, Hernández-Ruiz J, Cano A, Reiter RJ. Melatonin and Carbohydrate Metabolism in Plant Cells. Plants (Basel). 2021 Sep 15;10(9):1917. doi: 10.3390/plants10091917. PMID: 34579448; PMCID: PMC8472256.
Lines 381-385: The authors briefly mentioned melatonin's role in mitigating waterlogging stress, it does not adequately explore its effects on heavy metal stress.
A: In order to address the effect of melatonin on the alleviation of the stress induced by heavy metals we added a detailed explanation of the paper by Lv, Y., Zheng, Y., Wang, J. et al. Foliar application of melatonin reduces cadmium uptake and Cd-induced oxidative stress in tobacco under Cd stress. Plant Growth Regul (2025). https://doi.org/10.1007/s10725-024-01262-7. Line 174-194
Lines 361-367: The discussion on fungal communities focuses primarily on Ascomycota, neglecting other phyla like Basidiomycota and Mortierellomycota.
A: We are sorry, but the Authors of the cited paper did not provide any other information in the paper
Round 2
Reviewer 2 Report
Comments and Suggestions for Authors
The author did not complete the modification as required, and there are still large paragraphs describing a complete research work of others. We don't even need to look hard to see this. eg. line 264, Recently, Jofre et al. [34] explored...; line 277, Recently, Kwon et al., [21] selected...; line 306, In another study...; line 375, Emerging research suggests...; line 389, Madigan et al. [41] studied...; line 401, Additional research explored...; line 462, A very recent study [45] investigated...
If someone just list other people's research in this form, then he or she can almost mass-produce similar reviews by using AI to summarize. Therefore, I recommend that the author stick to the topic of melatonin and try to avoid the expression of what others have studied.
In addition, the word "in conclusion" is not recommended in line 515. Because there will be a chapter section that summarizes the whole text.
Author Response
The author did not complete the modification as required, and there are still large paragraphs describing a complete research work of others. We don't even need to look hard to see this. eg. line 264, Recently, Jofre et al. [34] explored...; line 277, Recently, Kwon et al., [21] selected...; line 306, In another study...; line 375, Emerging research suggests...; line 389, Madigan et al. [41] studied...; line 401, Additional research explored...; line 462, A very recent study [45] investigated...If someone just list other people's research in this form, then he or she can almost mass-produce similar reviews by using AI to summarize. Therefore, I recommend that the author stick to the topic of melatonin and try to avoid the expression of what others have studied.
A: In this review article, as correctly stated by the referee, "there are still large paragraphs describing a complete research work of others". We believe that discussing and summarizing the work of others is what review articles aim to do. This review article was produced by the two of us without any AI and we believe that it reflects the current state of this field.
In addition, the word "in conclusion" is not recommended in line 515. Because there will be a chapter section that summarizes the whole text.
A: done, as suggested by the reviewer.
Reviewer 4 Report
Comments and Suggestions for Authors
The authors have carefully addressed all comments. Accept in the present form.
Author Response
We thank the reviewer for the time spent on our manuscript and for the valuable suggestions.